# Mass-Synthesized Solution-Processable Polyimide Gate Dielectrics for Electrically Stable Operating OFETs and Integrated Circuits

**DOI:** 10.3390/polym13213715

**Published:** 2021-10-28

**Authors:** Rixuan Wang, Joonjung Lee, Jisu Hong, Hyeok-jin Kwon, Heqing Ye, Juhyun Park, Chan Eon Park, Joon Ho Kim, Hyun Ho Choi, Kyuyoung Eom, Se Hyun Kim

**Affiliations:** 1School of Chemical Engineering, Yeungnam University, Gyeongsan 38541, Korea; nl910213@gmail.com (R.W.); yeheqing5420@gmail.com (H.Y.); joon@ynu.ac.kr (J.H.K.); 2Department of Advanced Organic Materials, Yeungnam University, Gyeongsan 38541, Korea; joon@utp.or.kr; 3Research Institute for Green Energy Convergence Techonology, Gyeongsang National University, Jinju 52828, Korea; jisu225@postech.ac.kr; 4Department of Chemical Engineering, Pohang University of Science and Technology, Pohang 37673, Korea; hj1370@postech.ac.kr (H.-j.K.); cep@postech.ac.kr (C.E.P.); 5COMEC Corporation, Pyeongtaek 17957, Korea; komec66@daum.net; 6Department of Materials Engineering and Convergence Technology, Gyeongsang National University, Jinju 52828, Korea

**Keywords:** organic field-effect transistor, gate dielectric, polyimide, integrated logic gates

## Abstract

Polyimides (PIs) are widely utilized polymeric materials for high-temperature plastics, adhesives, dielectrics, nonlinear optical materials, flexible hard-coating films, and substrates for flexible electronics. PIs can be facilely mass-produced through factory methods, so the industrial application value is limitless. Herein, we synthesized a typical poly(amic acid) (PAA) precursor-based solution through an industrialized reactor for mass production and applied the prepared solution to form thin films of PI using thermal imidization. The deposited PI thin films were successfully applied as gate dielectrics for organic field-effect transistors (OFETs). The PI layers showed suitable characteristics for dielectrics, such as a smooth surface, low leakage current density, uniform dielectric constant (k) values regardless of frequency, and compatibility with organic semiconductors. Utilizing this PI layer, we were able to fabricate electrically stable operated OFETs, which exhibited a threshold voltage shift lower than 1 V under bias-stress conditions and a field-effect mobility of 4.29 cm^2^ V^−1^ s^−1^. Moreover, integrated logic gates were manufactured using these well-operated OFETs and displayed suitable operation behavior.

## 1. Introduction

Polymeric dielectrics are regarded as key components in the realization of organic electronics for the fabrication of scalable device arrays and integrated circuits [1,2]. Compared with conventional inorganic-based dielectric materials employed in the modern electronic industry, polymeric dielectrics are capable of providing excellent mechanical flexibility, good solution processability, and better compatibility with various types of substrates and organic channel materials [3,4]. Additionally, interface engineering with polymeric dielectrics, such as surface treatment on inorganic insulators, can offer potential capabilities for optimizing the molecular arrangements of organic semiconductors and trap-free interfaces for efficient charge transport, [2,5,6] influencing their compatibility with microfabrication technologies for highly integrated practical applications [5,7].

Indeed, various polymeric materials, including poly(methyl methacrylate), [8,9] polystyrene, [10] polyimide (PI), [7] and poly(4-vinylphenol) [11] have been utilized as gate dielectrics in organic field-effect transistors (OFETs) and integrated circuits. Among them, PIs are flexible, lightweight, endurable to thermal and chemical stress, [12] and have become promising candidates as gate dielectrics with good feasibility in the manufacture of organic electronics [7,13]. In particular, various types of PIs can be produced by the addition of various types of monomers and polymerization systems [14]. The first polyimide of significant commercial importance ‘Kapton’ was pioneered in the 1950s by workers at Dupont, who developed a successful route for the synthesis of high-molecular-weight PI involving a soluble polymer precursor (poly(amic acid) (PAA)) [15]. To date, this route continues to be the main route for the production of most PIs, enabling mass production. In this respect, PIs with these systems can be considered as the most promising materials for realizing a solution-processable organic electronic industry.

In this work, we synthesized mass-produced solution-processable conventional PAA (soluble PI precursor) through the factory method, and fabricated PI thin films through the imidization of PAA solution for polymeric dielectric layers of OFETs. To confirm the suitability of the fabricated PI films as gate dielectrics of OFETs, various characteristics, including surface roughness, dielectric constant (k), and leakage current properties were identified. Through these analyses, our fabricated PI films could reliably compensate for the electrically stable operation of OFETs under bias stress, showing a threshold voltage (V_th_) shift of less than 1 V. In addition, we demonstrated the potential of our coated PI layers in practical gate dielectrics for organic electronics by manufacturing integrated logic circuits. We expect that this study shows that many types of PIs reported so far have the potential to be industrially produced and used in device fabrication, and suggests a future direction for practical organic electronics in industrialization. 

## 2. Materials and Methods

### 2.1. Preparation of Materials

The precursor of PI (PAA) was synthesized with pyromellitic dianhydride (PMDA, TCI Chem/Japan) and 4,4′-diaminodiphenylether (ODA, TCI Chem./Japan) following the conventional PAA synthesis method, as shown in Figure 1a. For the synthesis of PAA, ODA and PMDA were dissolved in DMAc (TCI Chem./Japan) under dry nitrogen in an industrialized mass reactor (Appendix A) with a molar ratio of 1.000–1.005. The mixed solution was stirred at room temperature (~20 °C) for 16 h. 2,9-didecyldinaphtho [2,3-b:2’,3’-f] thieno [3,2-b] thiophene (C_10_-DNTT) was synthesized using previously reported methods [16].

### 2.2. Device Fabrication

Highly doped *n*-type Si wafers (resistivity < 0.005 Ω cm) with thermally deposited 300 nm SiO_2_ (Si/SiO_2_ wafers) produced from Namkang Hi-tech (Republic of Korea) and glass wafers (Eagle-XG) produced from iNexus. Inc. (Republic of Korea) were utilized as substrates for the material characterization and device fabrication. The wafers were clipped to a size of 2 × 2 cm^2^. Then, the wafers were cleaned with boiled acetone for 20 min and ultrasonically cleaned with acetone and isopropyl alcohol (IPA, Dasjung Chem./ Republic of Korea), each for 30 min. After these cleaning processes, the wafers were rinsed with acetone (J. T. Baker/US) and blown with nitrogen gas. To further eliminate residual contaminants on the surface, a UV–ozone cleaning process was conducted for 20 min. With these prepared Si wafers, the synthesized PAA precursor was deposited through a spin-coating process (2000 rpm for 30 s). For the imidization of PAA, thermal annealing was conducted at 200 °C, as shown in Figure 1b.

For the fabrication of OFETs and integrated circuits, 30 nm-thick Al gate electrodes were initially deposited on the prepared glass substrates using a thermal evaporator (deposition rate = 20 Å s^−1^; vacuum pressure = 10^−6^ Torr; substrate temperature = 25 °C). The synthesized PI layers were deposited using a previously mentioned method. Then, the fabrication of the OFETs was completed by the sequential deposition of a 50 nm organic semiconductor (C_10_-DNTT, deposition rate = 0.1–0.2 Å s^−1^; vacuum pressure = 10^−6^ Torr; substrate temperature = 25 °C) and 50 nm Au source/drain electrodes (deposition rate = 20 Å s^−1^; vacuum pressure = 10^−6^ Torr; substrate temperature = 25 °C).

### 2.3. Characterizations

The gel permeation chromatography (GPC) of the prepared solution was identified using an ACME9000/RI Detector (SHODEX KD-806/Japan/2 0.5 mL/min, 45 °C). The chemical functionalities of the PI films were identified by Fourier transform infrared spectroscopy (FTIR) using a Vertex 70 v (Bruker/Germany). The surface morphologies of the dielectrics and organic semiconductors were characterized using atomic force microscopy (AFM, Bruker Multimode 8/Germany) and by collecting two-dimensional grazing-incidence X-ray diffraction (2D-GIXD) data at the 9A beamline of the Pohang Accelerator Laboratory (PAL) in Korea. The electrical characteristics of the capacitor, OFET, and integrated circuits were obtained using an LCR meter (Agilent 4284A Precision LCR Meter, Keysight/US) and Keithley 4200 SCS parameter analyzer (Tektronix, Inc./US) under ambient air conditions.

## 3. Results and Discussion

The PAA solutions were synthesized by the polymerization of PMDA/ODA with DMAc solvents, as shown in Figure 1a. PAA solution was prepared by reacting with ODA and PMDA, which dissolved in DMAc under dry nitrogen in an industrialized mass reactor overnight. After the polymerization of PAA, the molecular weight was determined by GPC analysis. As shown in Appendix A, the prepared PAA had a molecular weight (Mw) of 231 kDa and a molecular weight distribution of 5.11 (Appendix A). With this prepared PAA solution, we fabricated PI layers following the method described in the Materials and Methods section and Figure 1b. The FTIR measurements were performed to confirm that the PI film was well formed. According to the FTIR spectra in Figure 1, the PAA film showed broad peaks around 1650 cm^−1^, which indicates the characteristic of amide C=O stretching modes [17]. These broad peaks disappeared in the spectra of the membrane after thermal imidization and became sharp peaks. The peaks at 1775 cm^−1^, 1720 cm^−1^, 1380 cm^−1^, and 738 cm^−1^ correspond to C=O asymmetric stretching, symmetric stretching, CN stretching, and C=O bending, which are primary indications of PI absorption bands, showing a larger transition with no characteristic PAA peaks [18].

Before utilizing the prepared PI films for electronic applications, we analyzed the film properties. Figure 2a shows AFM images of the prepared PI layers. The AFM image exhibited a smooth amorphous surface with a root-mean-square (RMS) roughness value of 0.212 nm, and this low RMS roughness value of less than 0.5 nm can provide a favorable environment for the growth of organic semiconducting crystals [5,19,20]. Furthermore, in order to identify the possibility of using this PI thin film as a dielectric layer for OFETs, the electrical characteristics were evaluated by fabricating a metal–insulator–metal (MIM)-structured capacitor with Al and thermally prepared PI layers (thickness: 150 nm) on Si/SiO_2_ wafers. As illustrated in Figure 2b, the dielectric constant (k) values of the PI were yielded at the level of 2.6, the value of which was roughly similar over a wide range of operating frequencies, from 2.62 at 70 Hz to 2.57 at 1 MHz. This phenomenon of showing constant k values according to the frequency means that there is no irregular arrangement of dipoles in the dielectric layers, which suggests the possibility of implementing a device without hysteresis by dielectrics when applied to a logic device including a transistor [20,21]. Figure 2c shows the leakage current density (I_leak_) of the PI films. I_leak_ was 1.98 × 10^−9^ A cm^−2^ at 2 MV cm^−1^ which is comparable or superior to that of other polymeric insulating materials [2,20]. Therefore, our synthesized PI layers have good insulating properties and suitable characteristics for gate dielectrics of OFETs or integrated circuits.

Generally, the morphology of organic semiconducting layers deposited on the polymeric layers (PI layers in our work) is a significant factor in configured device applications, because charge-carrier transport in OFETs with bottom-gate top-contact configuration mainly occurs in a few-nanometers-wide channel at the interface between organic semiconductors and gate dielectrics [22]. Indeed, the crystalline morphology of the active layers (organic semiconducting layers), which are determined by the surface properties of the gate dielectric layer (surface energy, surface roughness, and chemical functionality), strongly affect the electrical performance of OFETs [23,24]. Thus, the morphologies of the organic semiconductors on the PI layers were analyzed.

Figure 3a shows the surface topological image of C_10_-DNTT on the PI layers. The C_10_-DNTT on our PI layers showed the formation of mixed crystals containing smooth grains observed in wide areas and sharp needle-like grains protruding on the surface with grain sizes of hundreds of nanometers, as previously reported [25,26]. To further understand the orientation of C_10_-DNTT, 2D-GIXD measurements of these layers were taken. As shown in Figure 3b, the pattern image represents the mixed molecular orientation of C_10_-DNTT with peaks related to face-on orientation (white asterisks) and edge-on orientation (white dotted line with 00l, 11l, 02l, and 12l peaks) [27], which is consistent with the AFM results. It was confirmed that this type of C_10_-DNTT morphology showed suitable charge transfer for OFETs, as in previous studies [20]; therefore, it was found that our PI dielectric thin film also gave rise to an organic semiconducting shape suitable for manufacturing OFETs and integrated circuits.

The fabrication of bottom-gate top-contact OFETs was conducted according to the method described in the Materials and Methods section (Figure 4a). Manufactured OFETs operated in a saturation regime with a gate voltage (*V_G_*) from 20 to −20 V and source-drain voltage (*V_D_*) of −20 V for transfer characteristics under ambient air conditions. The electrical parameters of the devices were calculated using the following equation: (1)$$I_{D}=\mu_{FET}C_{i}\frac{W}{2L}{\left(V_{G}-V_{th}\right)}^{2}$$
where *C*_i_ is the capacitance per unit area (measured value was 20 nF cm^−2^ in same geometry of MIM capacitor on glass wafers), W/*L* is the ratio between the channel width (1500 μm) and length (150 μm) equivalent to 10, and I_D_ and *V*_th_ are the drain current and threshold voltage of the devices, respectively.

Figure 4b shows the transfer and output characteristics of OFETs with PI dielectrics. According to the transfer characteristics, OFETs showed a typical *p*-type operation behavior with an average field-effect mobility (*μ_FET_*) of 4.29 cm^2^ V^−1^ s^−1^, an ON/OFF ratio of 1.3 × 10^6^, and a *V*_th_ of −1.56 V. In particular, the transfer curve exhibited a low hysteresis shape, indicating that, in addition to the aforementioned dipole disordered effect, there is almost no trapping phenomenon occurring at the interface between the channel and dielectric layers. In addition, the output characteristics in Figure 4c display obvious linear and saturation regions, which indicate a reliable operation of the transistors. In addition, the saturation current in the transfer curve at a *V_G_* of −20 V is similar to the saturation current shown in the output curve at *V_G_* of −20 V. Furthermore, the bias stress stability data suggested highly stable electrically operating features of PI dielectric-based OFETs under ambient air conditions (20 °C and 50% relative humidity), as shown in Figure 4d. Indeed, the shift in *V*_th_ during the bias stress test was less than 1 V even at a driving voltage of 20 V, which is comparable to the stable polymeric dielectrics reported previously [28,29].

The feasibility of our synthesized PI dielectrics for application in integrated circuits was further investigated by assembling *p*-type OFETs for the integration of *p*-type integrated logic gates [30,31]. Figure 5a illustrates the circuit diagrams and top-view OM images of the logic gates. Our NOT gates were designed by connecting two types of OFETs in series: diode-connected OFETs for load transistors and typical *p*-type OFETs for drive transistors. The W/L ratios of the OFETs were 2 (W: 100 μm, L: 200 μm) and 20 (W: 100 μm, L: 2000 μm) for the load and drive transistors, respectively. Figure 5b shows the operation behavior of our NOT gates with voltage transfer characteristics (VTC) and voltage gain curves depending on the supply biases (V_DD_) of 10, 15, and 20 V. According to the VTC curve, the output voltage (V_OUT_) was extracted as values close to V_DD_ under low input voltage (V_IN_) conditions and values close to zero under high V_IN_ conditions. This was induced by the typical *p*-type NOT gate operation [32]. A high V_IN_ bias condition creates a low current characteristic because the gate-source voltage of the driving transistor is small; therefore, the load transistor is strongly turned on. Thus, V_OUT_ is pulled down to zero. In contrast, a low V_IN_ bias condition leads to similar values between the gate and source voltages of both the drive and load transistors. In this situation, the dimension of the drive transistor’s W/L ratio was significantly larger than that of the load transistor, and the V_OUT_ was pulled up toward the V_DD._

With this operation mechanism, the “0” and “1” signals of the bias can be named as low and high voltages, respectively. The output of the NOT gates was the opposite of the input/output conditions, such that “1” of the output voltage signal was extracted from “0” of the input voltage signal or “0” of the output voltage signal was extracted from “1” of the input voltage signal. The gain values of these NOT gates (dV_OUT_/dV_IN_) were extracted from the VTC curve, showing maximum values of 0.92, 1.14, and 1.27 for 10, 15, and 20 V, respectively. These driving results are similar to those of typical NOT gate operation, indicating that our PI-based OFETs were successfully applied to NOT gates.

More complex integrated circuit NAND and NOR gates were fabricated by one load OFET and two drive OFETs, as illustrated in Figure 5a [30]. In the case of NAND gates, two *p*-type drive OFETs were connected in parallel with each other. Through these manufactured NAND gates, we observed how the output signal was driven according to various logic input signals (see the NAND output signal in Figure 5c). When both of the inputs were in logic low states, such that V_A_ and V_B_ were equal to 0 V, both of the OFETs were switched on, resulting in a direct path between V_DD_ and V_OUT_. Therefore, the output signal was a logic high state. Similarly, the input condition when one of the inputs was logic high and the other was logic low operated one of the two transistors connected in parallel, and connected V_DD_ and V_OUT_ directly. Thus, the output signal was a logic high state. On the other hand, when both inputs were in logic high states (40 V of input bias), none of the transistors connected in parallel operated, causing ground (GND) to connect directly to V_OUT_. Thus, the output signal of the NAND gates was a logic low state.

For the NOR gates, two *p*-type drive OFETs were connected in series with each other. When both of the inputs were in a logic low state of 0 V input bias at V_A_ and V_B_, series-connected OFETs were switched on, resulting in the creation of a direct path between V_DD_ and V_OUT_ (see the NOR output signal in Figure 5c). In this situation, the output signal would be a logic high state. On the other hand, when any of the inputs were logic high (~40 V), one or more drive transistors were switched off, which triggered disconnection between V_DD_ and V_OUT_. Thus, the output signal was logically low. This operation coincided with the conventional logic gate drive, which indicates that our synthesized PI-based OFETs were successfully applied to practical integrated circuits. However, the voltage level of these logic gates was relatively high (~40 V), and this point can be a disturbing part for the industrialization of OFETs base PI layers. However, it can be improved if device geometry is improved, such as by applying short channel devices through photolithographic processes usually used in industry and high-k insulating materials that together can maintain high capacitance values [5,33]. Considering the results, this study showed the driving stability of OFETs with PI gate insulators, which were fabricated through factory methods. Additionally, this study suggests the possibility that many types of PIs reported so far, even if they are industrially produced, can be used in device fabrication, and suggests a future direction (e.g., improving the device geometry for real industrialized device manufacturing).

## 4. Conclusions

In this work, we synthesized a conventional PAA precursor with an industrialized reactor for mass production using a previously reported conventional PI synthesis method, and a synthesized PAA solution was successfully applied to coat a PI thin film via thermal imidization. The fabricated PI thin films showed suitable characteristics for the dielectrics of OFETs, such as a smooth surface, low leakage current density, constant k values regardless of frequency, and compatibility with organic semiconductors. Considering these characteristics, we were able to fabricate well-operated OFETs using these PI layers, which exhibited V_th_ shifts lower than 1 V in a bias stress environment and a *μ_FET_* of 4.29 cm^2^ V^−1^ s^−1^. In addition, these well-operated OFETs were successfully utilized for fabricating integrated logic gates, and they exhibited suitable operating properties. Therefore, we conclude that our mass-produced PAA/PI materials have potential for the practical industrialization of organic electronics in the near future. Additionally, we believe that our work suggests the possibility that many types of PIs reported so far, even if they are industrially produced, can be used in device fabrication, and suggest a future direction.

## Data Availability

Not applicable.

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
