# Peer review of "Mass-Synthesized Solution-Processable Polyimide Gate Dielectrics for Electrically Stable Operating OFETs and Integrated Circuits"

_polymers, 2021, doi:10.3390/polym13213715_

Round 1

Reviewer 1 Report

Dear Authors,

I really appreciated to read your manuscript: it is well structured, well understandable, results and figures well described and  flawless, conclusions are well reported.

I did not find any reasonable point of costructive discussion because I did not find any hole in the text of the manuscript.

In my own opinion is a great step forward in organic semiconductor, and in special mode, as the Authors made, in organic LED.

My best regards.

Reviewer 2 Report

In this work, Wang et al. demonstrate that it is possible to apply an industrial process for the production of polyimide (PI) thin films to the fabrication of OFETs where PI films act as gate dielectrics.

The overall quality of the work is good. The applied methodology is rigorous and correct, and the results are clearly presented and discussed. 

My biggest concern is about the novelty of the work. Authors should, for instance, highlight the progress made with respect to other works previously reported in the literature focused on OFETs fabricated with PI-films gate dielectrics (see Zou et al,  "Polyimide-based gate dielectrics for high-perfortmance organic thin-film transistors", Journal of Material Chemistry C, 2019, 7, 7454-7459). 

Moreover, the paper should be better organized in some points, and authors should add some missing info on the fabricated devices, which are essential to compare the obtained results to the existing literature. I'll be more precise in the bullet-point list of comments.

By the way, in my opinion, the paper is suitable for publication in Polymers journal after minor revision, provided that the authors have addressed some  issues:

1) All acronyms should be defined the first time they appear in the paper (e.g. PAA in the abstract). Authors should carefully check this. Also, all the symbols indicating parameters or variables should be defined (for instance, the definition of the symbol indicating the charge carrrier mobility).

2) Authors refer to two schemes (Scheme 1 and Scheme 2) which are however absent from the main paper. They also refer to a figure (S1) probably enclosed in a supplementary material which is not provided. Please check.

3) Some paragraphs in section 3 refer to applied methodologies and methods, and should be moved to Section 2. For instance, lines 109-115 (page 3) are a mere repetition of what already written in Section 2.1, so I invite the authors to remove lines 109-115 (page 3) and to merge them with Section 2.1. Another example: lines 169-172 should be moved to Section 2.2, as well as the figure 4(a) showing the schematic of the fabricated OFET device.

4) All the details about the geometric features of the fabricated OFET device are missing, and should be added, for instance: Au source/drain contacts lateral size, channel length and with (only the ratio is reported, but it's not enough), and most of all the thicknesses of both the C10-DNTT and the PI dielectric layers! Also, has a standard photolithographic process been used? Or not? All these details are indeed essential to help the reader evaluating the performance of the OFET device introduced by the authors.

5)  Logic voltage levels (40 V) of the fabricated devices are significantly high and not compatible to practical industrialization, as claimed by the authors in the conclusion. However, it obviously depends on the geometry of the device, because the electric field strength (which is the true driving force)  depends of course on the distance between the points of voltage application. Therefore, I strongly recommend the authors to add info on the OFET geometric features, as well as to comment on the high voltage levels of the implemented logic with respect to other OFET-based technologies reporting lower voltage values (see for instance Lai et al., "All-Organic, Low Voltage, Transparent and CompliantOrganic Field-Effect Transistor Fabricated by Means of Large-Area, Cost-Effective Techniques", Applied Science, 2020, 10, 6656).
